# Introspective access to value-based multi-attribute choice processes

Adam Morris [1] ✉, Ryan W. Carlson[2], Hedy Kober [3,4] & M. J. Crockett[1,4] ✉

People routinely choose between options varying on multiple attributes – homes to rent, movies to watch, and so on. Here, we test how much awareness people have of the mental processes underlying these choices. We develop a method to quantify awareness of value-based multi-attribute choice processes that accounts for diverse choice strategies. Across five studies, participants make choices and then report how they believe they made them. We use computational modeling to identify the process revealed in their choices, and compare it to their self-reports to quantify individuals' accuracy about their choice process. While we observe substantial variation in accuracy, participants are often highly accurate about their choice process – more accurate than predicted by a sample of decision scientists – and more accurate than informed third-party observers, suggesting evidence for introspection. These results challenge notions that we are strangers to ourselves and instead suggest that people often know how they made value-based choices.

A recurring question in the cognitive and behavioral sciences is how much introspective access people have to mental processes underlying their choices. Some mental events, such as a song playing in your head or the steps of mental arithmetic, produce conscious internal experiences about which people can report and reason[1]—while others, such as visual computations in V1, occur unconsciously and cannot be directly accessed or reported[2]. In what ways are choice processes more like the former versus the latter? This question has profound implications—both for the basic nature of decision-making[3], and for applied considerations such as how to improve people's choices[4] or how to attribute responsibility and blame[5–7].

Curiously, this question has received very different answers in different parts of psychology. As posited in the seminal paper of Nisbett and Wilson[8], one possibility is that people are largely unaware of the factors and mental operations underlying most of their choices[9–16], and that people's explanations for their own behavior are largely post-hoc inferences or confabulations[17–20]. This idea has been widely influential[21–23]—motivating, for instance, the development of nudge-style interventions that prioritize altering external choice architectures (rather than helping people control and improve their own choice process)[24–26]. In contrast, other areas of psychology have posited that

people are, or can become, aware of many key processes underlying their choices[27]. For instance, a widespread assumption in clinical psychology is that people can become aware of mental processes underlying dysfunctional decisions[28–30], and this assumption is at the heart of many effective therapeutic interventions that prioritize introspective insight[31–35]. Moreover, people's ratings of confidence in their choices often accurately track the quality of those choices[36–39], suggesting that they might have awareness of at least some aspects of their choice processes[40].

Here, we rigorously examine people's awareness of their choice processes in one ubiquitous kind of choice: value-guided, multi-attribute choice. This kind of choice is where people seek to satisfy their preferences by weighing together multiple attributes—for instance, deciding what movie to watch by weighing together the genre, quality of the acting, number of awards won, etc.; or deciding what lodging to rent for your summer vacation by weighing the home's location, size, decor, etc. These choices constitute much of our everyday decision-making[41–45]. Yet, despite their ubiquity, there is surprisingly little research directly testing people's awareness of the mental processes underlying these decisions[3,46,47]. Some research has touched on the question indirectly, e.g., by testing whether people's

[1]Department of Psychology, Princeton University, Princeton, NJ, USA. [2]Booth School of Business, University of Chicago, Chicago, IL, USA. [3]Department of Psychology, University of California: Berkeley, Berkeley, CA, USA. [4]These authors jointly supervised this work: Hedy Kober, M. J. Crockett. ✉e-mail: am9578@princeton.edu; mj.crockett@princeton.edu

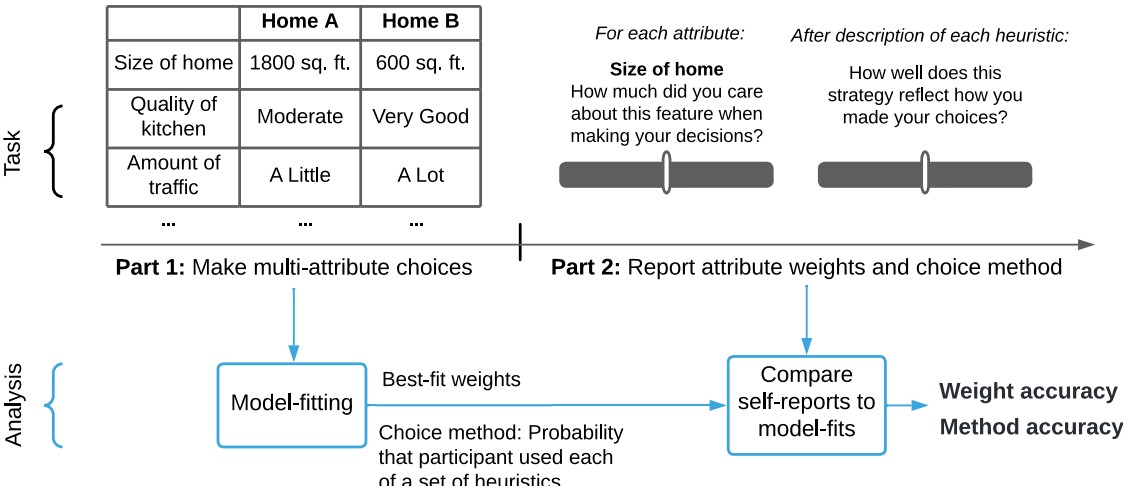

**Fig. 1 | Overview of the Awareness of Choice Processes (ACP) task.** In Part 1, participants made two-alternative forced choices between multi-attribute options (e.g., homes to rent, or movies to watch). On each of 100 choice trials, participants chose between two options, each characterized by nine attributes that were experimentally manipulated on each trial. Then, in Part 2, participants reported how they believe they made their choices in Part 1 (on 100-point scales). Finally, we quantified individuals' accuracy about their choice process by comparing their self-reports to the attribute weights and choice method identified via computational modeling (see Methods for details).

confidence ratings accurately track their decision quality in these contexts[36,37,40], or whether directing people to introspect changes their decision quality[48–50]. The empirical work that has tested for awareness directly in these choice contexts has yielded mixed results[47,51–57], with much of it criticized on methodological grounds[3,58,59]. For instance, prior work from the 1970s–1980s testing awareness in value-guided, multi-attribute choice incorrectly assumed that all people use the same "classically rational", expected-value-maximization choice process[8,15,51,56,57,60,61], when in fact people often use heuristic methods to simplify their choices[12,44,62–64]; this methodology likely mischaracterized the extent of people's awareness[63,65–67] (see Supplementary Information [SI] 1.5 for further discussion). In addition, prior work mostly tested small, convenience samples of undergraduates, and did not include incentivized choices.

Overcoming past methodological limitations, we exploit advances in computational modeling of choice processes to rigorously quantify and examine people's awareness of the mental processes underlying their value-guided, multi-attribute choices. As mentioned above, in these choice contexts people typically assign weights to each attribute, representing how much they value that attribute (e.g., how much size matters to them when choosing a home)[41,42,68]. Moreover, people employ a variety of methods to convert those weights into a choice; as discussed above, people sometimes do full expected-value maximization[41,42], but there are also heuristics (e.g., "take the best" heuristics that focus on just one attribute) that people often use to simplify the choice problem[12,44,62–64].

We leveraged this past research to create our Awareness of Choice Processes (ACP) task (Fig. 1). In our studies, participants made value-guided decisions between options that were experimentally manipulated on multiple attributes, and then reported how they believed they made those decisions: the weights they believed they placed on each attribute, and the choice method they believed they used. To report their choice method, participants read intuitive descriptions of three common heuristics and reported whether or not they used each heuristic (as opposed to the "classically rational" expected-value method). Then, in our analysis, we identify the weights and choice method revealed in participants' actual choices by fitting established computational models to those choices. Finally, we compare these model-fitting results to participants' self-reports to quantify each individual's accuracy about their attribute weights and choice method.

Critically, if participants can accurately report aspects of the choice process they used, this does not necessarily imply that they have direct introspective awareness of that choice process; participants could be employing other mechanisms, such as making inferences from observing their own choices[17–19,69], or relying on general background knowledge about choice[8]. (Intuitively, the difference between direct introspective awareness and non-introspective inference is the difference between hearing a song playing in your head versus inferring that a song is playing in your head because you've been humming it.) To adjudicate between these possibilities, we also developed an "observer" version of the ACP task. In this version, an independent set of observer participants were matched with original "decider" participants from the ACP task, shown all the deciders' choices, and asked to report the deciders' attribute weights and choice method (with monetary incentives for accuracy). Since observers have inferential but not introspective evidence about the deciders' choice process, if observers can match deciders' accuracy about that choice process, this would suggest that deciders' accuracy is explainable by inferential mechanisms. But if deciders are more accurate than observers about their own choice process, this would suggest that participants' accuracy comes partly from privileged, first-person awareness[8].

Across five studies (total $N = 1144$; online samples of U.S. participants nationally representative for age, race, and gender), we used the ACP task to characterize the extent and mechanisms of awareness in multi-attribute choice. We first examined participants' accuracy in reporting the attribute weights and choice methods they used across two choice domains: hypothetical homes to rent (Studies 1A, 1B), and real movie trailers to watch (Study 2). Then, we recruited observers who were matched to the decider participants from the home rental variant (Study 3A) and the movie variant (Study 3B), and assessed their accuracy in order to probe the mechanisms underlying deciders' accurate reporting. Finally, to test how participants' accuracy compared to the expectations of experts in the field, we asked a sample of decision scientists to predict participants' level of accuracy in our paradigm (Study 4), and compared these predictions to the actual accuracy observed in our experiments. Study 4 provides an empirical benchmark of the field's prior expectations about choice process awareness, against which we can compare our experimental results.

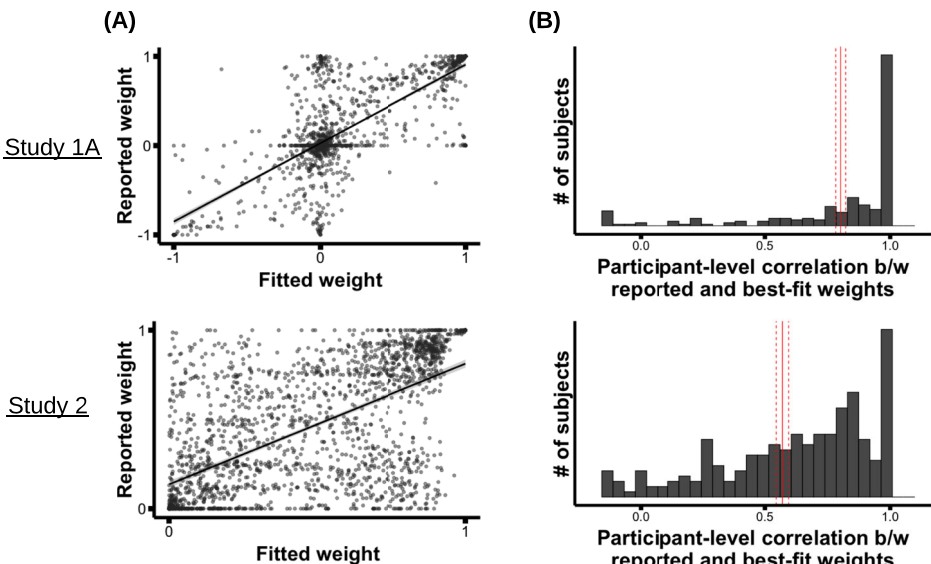

**Fig. 2 | Observed levels of weight accuracy. A** Overall relationship between reported and best-fit attribute weights, in Study 1A and 2. Each dot represents one attribute from one participant; the least-squares line is shown (with a 95% confidence interval band around it). **B** Histogram of participant-level correlations between reported and best-fit weights; red line shows the mean, with the dashed lines around it showing the standard error of the mean (SEM). For all top plots (Study 1A), $N = 237$; for all bottom plots (Study 2), $N = 235$. (The plots for Study 1B are similar to Study 1A, and are presented in SI 3.3.).

## Results

In Studies 1A-1B and 2, we recruited 300 Prolific participants per study to complete the ACP task (Fig. 1). The studies differed only in the type of choice trials in Part 1. In Studies 1A-1B, participants made value-based choices between hypothetical homes to rent, which varied on attributes such as size, kitchen quality, and amount of traffic nearby (see SI 1.1 for full attribute list). The values of these attributes were manipulated on each choice trial. Study 1A used a fixed set of pre-fabricated choice trials shown to all participants; Study 1B generated a unique set of choice trials for each participant, to ensure that the results from Study 1A were not an artifact of its choice set.

Study 2 tested whether the results generalized to a different choice domain with non-hypothetical choices. In Study 2, the choices were between real movies, which varied on attributes such as quality of the plot, dialog, and cinematography. These attributes were drawn from a database aggregating user ratings of movies across different dimensions[70-72]. At the end of Study 2, participants had to watch the trailer of one of the movies they chose, making their choices incentivized. In all studies, after completing the ACP task, participants completed a battery of individual difference measures discussed below.

To identify each participant's choice process, we fit a set of established computational models to their choices. We restricted the set of potential choice methods to either the "classically rational" model of multi-attribute choice (which places graded weights on each attribute and combines them linearly to compute option values[42]), or a combination of three common heuristics[41,62,73,74]. Each heuristic simplifies the value computation in a different way, and can be combined to form six different choice methods that are common alternatives to classical expected value maximization (see Methods).

Using these models, we extracted two key pieces of information about each participant's choice process: their set of best-fitting attribute weights, and the probability that they used each choice method (i.e., each set of heuristics). We confirmed in a variety of ways that our model-fitting procedure effectively extracted this information, including by simulating data from artificial decision-makers and verifying that the fitted models correctly identified their weights and choice method (see Methods). Notably, past work on awareness of multi-attribute choice processes assumed that all participants used the "classically rational" model; here we demonstrate empirically that this

model was used by only a minority of participants (40% across all studies), underscoring the importance of individual model-fitting for studying awareness of choice processes.

We then compared this model-extracted information about participants' choice processes to participants' self-reports about their choice processes. This allowed us to quantify two aspects of choice process awareness: participants' accuracy about their attribute weights (denoted "weight accuracy") and their choice method (denoted "method accuracy"). Since Studies 1A-1B and 2 yielded similar results, we present their results together.

### Weight accuracy

In all studies, participants' self-reported attribute weights correlated strongly with the best-fit weights (Fig. 2A). We estimated a linear mixed effects model regressing reported weights on best-fit weights, with random intercepts and slopes per participant and attribute. The relationship was significant (betas are standardized regression coefficients, which represent effect sizes, and all brackets indicate 95% confidence intervals; Study 1A: $\beta = 0.65$, $t(1.9) = 29$, $p = 0.002$, 95% CI for $\beta = [0.58, 0.69]$; Study 1B: $\beta = 0.74$, $t(5.4) = 29$, $p < 0.001$, 95% CI for $\beta = [0.69, 0.8]$; Study 2: $\beta = 0.52$, $t(12) = 14$, $p < 0.001$, 95% CI for $\beta = [0.44, 0.59]$). (The distributions of reported and best-fit weights for each attribute separately can be found in SI 3.4.)

As an individual-level measure of weight awareness, we then computed for each participant the Pearson correlation between their best-fit and reported weights (across the nine attributes). Since each correlation is computed over only nine attributes, the individual estimates are noisy; but analyzing those correlations in aggregate paints a picture of the distribution of individual-level weight accuracy (and is a commonly used measure in past work[51,75]).

Figure 2B shows a histogram of those participant-level correlations. These correlations had an average of $r = 0.80$ in Study 1A ($t(235) = 40$, $p < 0.001$, 95% CI for $\beta = [0.76, 0.84]$), 0.86 in Study 1B ($t(250) = 63$, $p < 0.001$, 95% CI for $\beta = [0.84, 0.89]$), and 0.57 in Study 2 ($t(233) = 23$, $p < 0.001$, 95% CI for $\beta = [0.52, 0.62]$). As an additional measure, we also computed, for each participant, the absolute difference between their reported and fitted weights for each of the nine attributes, and then averaged across attributes to obtain the participant's "absolute weight error". Participants showed similarly

high levels of weight accuracy by this measure: Their fitted and reported weights differed by an average of 0.13 [0.12, 0.15] in Study 1A, 0.12 [0.11, 0.14] in Study 1B, and 0.21 [0.20, 0.23] in Study 2. Intuitively, since the weight values range from −1 to 1, participants were accurate to within 1 part in 10 to 1 part in 20. In sum, participants reported their attribute weights with exceptional accuracy. This weight accuracy is significantly higher than what has been reported in most past work on value-guided multi-attribute choice; see SI 3.10 for discussion.

The mode in Fig. 2B of participants with a correlation close to 1.0 is due largely to participants who used only a single attribute; all they had to do to earn a correlation of 1.0 was report which attribute they were using, which most could do. (These participants made up around half of the sample in Studies 1A-1B, but only 15% of the sample in Study 2; see SI 3.4.) Excluding those participants, the overall relationship between participants' reported and best-fit weights was still strong (Study 1A: $\beta = 0.44$, $t(9.3) = 13$, $p < 0.001$, 95% CI for $\beta = [0.37, 0.52]$; Study 1B: $\beta = 0.58$, $t(6.8) = 25$, $p < 0.001$, 95% CI for $\beta = [0.53, 0.62]$; Study 2: $\beta = 0.43$, $t(10.0) = 12$, $p < 0.001$, 95% CI for $\beta = [0.35, 0.5]$), as was the average participant-level correlation (Study 1A: 0.75 [0.70–0.79]; Study 1B: 0.82 [0.80–0.85]; Study 2: 0.52 [0.47–0.57]). We also computed one additional measure of weight accuracy—the amount of information gained about a participant's best-fit weights from knowing their reported weights—that intrinsically favors participants who exhibited more complex patterns of weights, and again found high levels of accuracy: On average, participants' reported weights contained 79% [75–82%] of the total information about their fitted weights (Study 1A; Study 1B: 81% [78–84%]; Study 2: 72% [70–74%]; see SI 3.9 for details). Thus, the high observed weight accuracy was not merely due to participants who used only a single attribute.

Of course, even among participants using multiple attributes, many attributes are still assigned a weight of zero (or near zero); a concern is that participants' high weight accuracy could be due merely to knowledge of these ignored attributes, rather than a more nuanced awareness of the weights on the attributes they actually used. To rule out this possibility, we re-computed the measures of weight accuracy while dropping any attributes whose reported weight had an absolute value of less than 0.05. The results were similar to those reported above. The overall relationship between the reported and best-fit weights was still strong (Study 1A: $\beta = 0.43$, $t(914) = 19$, $p < 0.001$, 95% CI for $\beta = [0.39, 0.48]$; Study 1B: $\beta = 0.55$, $t(6.8) = 23$, $p < 0.001$, 95% CI for $\beta = [0.50, 0.61]$; Study 2: $\beta = 0.44$, $t(11.9) = 13$, $p < 0.001$, 95% CI for $\beta = [0.37, 0.51]$), as was the average participant-level correlation (Study 1A: 0.75 [0.69–0.8]; Study 1B: 0.80 [0.76–0.84]; Study 2: 0.51 [0.46–0.56]) and percent information contained in the reported weights (Study 1A: 77% [73–81%]; Study 1B: 78% [75–82%]; Study 2: 71% [69–74%]). We also re-computed these measures while only including each participants' top three highest-weighted attributes, and found similar results: The average correlation was 0.71 [0.62–0.79] (Study 1A; Study 1B: 0.80 [0.73–0.86]; Study 2: 0.42 [0.34–0.51]), and the percent information contained in the reported weights was 82% [78–86%] (Study 1A; Study 1B: 84% [80–87%]; Study 2: 76% [73–79%]). In sum, participants were on average quite accurate about the weights they had placed on all the attributes, including the ones to which they had assigned a high weight.

## Method accuracy

In addition to accuracy about attribute weights, we examined participants' accuracy in reporting the method they had used to convert those weights into a choice. Recall that the available choice methods were either the classical expected-value-maximization method[42], or a set of three common simplifying heuristics which could be combined to form six alternative choice methods[41]. Notably, our model-fitting procedure extracted two types of information about participants' choice method: the probability that a participant used each individual heuristic, and the probability that a participant used each potential combination of heuristics (which together form a single choice method). We leverage both types of information to paint a rich picture of participants' knowledge of their choice method.

To examine participants' accuracy about whether they used each individual heuristic, we first analyzed the relationship between the model-estimated probability that participants used each heuristic and the participants' self-reported extent to which they used each heuristic. Figure 3A shows this relationship, aggregating all heuristics. The relationship was significant (linear mixed effects models; Study 1A: $\beta = 0.22$, $t(210) = 5.9$, $p < 0.001$, 95% CI for $\beta = [0.14, 0.30]$; Study 1B: $\beta = 0.25$, $t(230) = 7.1$, $p < 0.001$, 95% CI for $\beta = [0.18, 0.32]$; Study 2: $\beta = 0.27$, $t(130) = 7.2$, $p < 0.001$, 95% CI for $\beta = [0.20, 0.35]$). The relationship was also significant for each heuristic alone (see SI 3.5).

To interpret the relationship between the reported and model-estimated probabilities of each heuristic, it is helpful to dichotomize each axis—i.e., binning participants into "using" or "not using" each heuristic. We will say as shorthand that a participant was more likely than not to have used a heuristic if the model-estimated probability of using it was >0.5, and that a participant reported being more likely than not to have used a heuristic if their reported extent of using it was >0.5 (i.e., we split each probability around its midpoint). Aggregating across all heuristics, participants who used a given heuristic were about twice as likely to report having used that heuristic, compared to participants who did not use it (Fig. 3B; logistic mixed effects models; Study 1A: $\beta = 0.95$, $z = 4.9$, $p < 0.001$, 95% CI for $\beta = [0.57, 1.3]$; Study 1B: $\beta = 1.2$, $z = 6.3$, $p < 0.001$, 95% CI for $\beta = [0.82, 1.6]$; Study 2: $\beta = 1.6$, $z = 5.8$, $p < 0.001$, 95% CI for $\beta = [1.0, 2.1]$).

These results demonstrate that participants—as a group—reported their heuristics far more accurately than chance. They do not, however, quantify any individual's accuracy about their heuristics. To quantify individual-level accuracy, we first calculated, for each participant, the absolute difference between their reported extent of using each heuristic and the model-estimated probability of that heuristic, averaged across all heuristics (denoted "heuristic error"). Participants showed significantly less heuristic error than chance (Fig. 3C; Study 1A: mean error = 0.34 [0.32, 0.36], chance = 0.42, difference $t(236) = −7$, $p < 0.001$, Cohen's $d = −0.47$, 95% CI for Cohen's $d = [−0.62, −0.33]$; Study 1B: mean = 0.33 [0.31, 0.35], chance = 0.42, $t(250) = −10$, $p < 0.001$, Cohen's $d = −0.62$, 95% CI for Cohen's $d = [−0.78, −0.47]$; Study 2: mean = 0.28 [0.26–0.3], chance = 0.43, $t(234) = −15$, $p < 0.001$, Cohen's $d = −1.0$, 95% CI for Cohen's $d = [−1.2, −0.85]$).

In addition to analyzing participants' accuracy about their use of each heuristic, we can also quantify their accuracy about their choice method as a whole—i.e., how accurate they were in reporting the overall combination of heuristics they did or did not use. To assess this, we combined participants' reported use of each heuristic to identify the overall choice method each participant believed they had been most likely to use. We then examined the model-estimated probability that the participant actually used that reported method. Specifically, for each participant, we divided the model-estimated probability of their reported choice method by the probability of the most likely method, yielding a Bayes factor of the reported method relative to the best-fitting method. This metric ranges from 0 to 1, where 1 indicates that the participant reported the best-fitting method, and lower numbers indicate that they reported worse-fitting methods. Participants on average reported models that fit their choices about half as well as the best-fitting model, which is much better than what could be achieved through random guessing (Study 1A: mean Bayes factor = 0.46 [0.40, 0.51], chance = 0.23, difference $t(236) = 7.8$, $p < 0.001$, Cohen's $d = 0.51$, 95% CI for Cohen's $d = [0.39, 0.63]$; Study 1B: mean = 0.45 [0.40, 0.51], chance = 0.23, $t(250) = 7.9$, $p < 0.001$, Cohen's $d = 0.50$, 95% CI for Cohen's $d = [0.38, 0.62]$; Study 2: mean = 0.56 [0.5–0.62], chance = 0.24, $t(234) = 11$, $p < 0.001$, Cohen's $d = 0.70$, 95% CI for Cohen's $d = [0.56, 0.85]$).

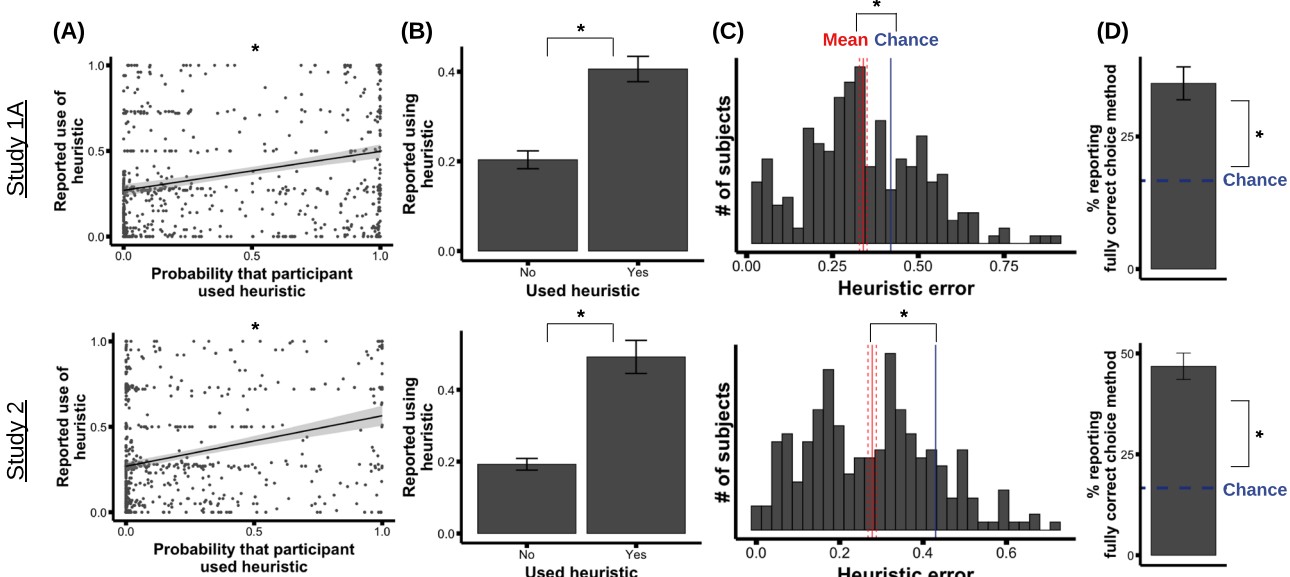

**Fig. 3 | Observed levels of method accuracy. A** Participants' reported extent of using each heuristic (y axis) plotted against the model-estimated probability that participants used that heuristic. Each dot is one heuristic for one participant; the least-squares trend line is plotted (with a 95% confidence interval band around it). Both slopes are significantly above zero ($p < 0.001$ for both plots). **B** Percentage of participants who reported a > 50% extent of using each heuristic (y axis), as a function of whether the model-fitting estimated a > 50% chance that they actually used that heuristic (x axis). Error bars indicate SEM. Participants were significantly more likely to report using a heuristic if they in fact were likely to be using it ($p < 0.001$ for both plots). **C** Histogram of participants' "heuristic error": the absolute difference between their reported extent of using each heuristic and the model-estimated probability of using that heuristic, averaged across heuristics. Red line shows the mean, with the dashed lines around it showing SEM; the dark blue line shows the heuristic error expected by chance. The average heuristic error was less than chance ($p < 0.001$ for both plots). **D** Percentage of participants who reported the fully correct choice method (i.e., correctly reported whether they used each heuristic or not); this percentage was significantly greater than chance ($p < 0.001$ for both plots). Error bars indicate SEM. In all plots, asterisks indicate significant effects at $p < 0.05$. For all top plots (Study 1A), $N = 237$; for all bottom plots (Study 2), $N = 235$. (The plots for Study 1B are similar to Study 1A, and are presented in SI 3.3.).

Finally, as a coarse but easily interpretable metric of overall method accuracy, we analyzed whether each participant reported the overall best-fitting choice method or not (Fig. 3D). Participants were far more likely than chance to report the fully correct method (chance was 1/6 because there were six possible choice methods; Study 1A: mean = 35% [29–41%], the difference from chance $X^2(1) = 56$, $p < 0.001$, Cohen's $h = 0.43$, 95% CI for Cohen's $h = [0.30, 0.54]$; Study 1B: 36% [30–42%], $X^2(1) = 65$, $p < 0.001$, Cohen's $h = 0.44$, 95% CI for Cohen's $h = [0.31, 0.55]$; Study 2: mean = 47% [40–53%], $X^2(1) = 152$, $p < 0.001$, Cohen's $h = 0.67$, 95% CI for Cohen's $h = [0.53, 0.80]$).

## Individual differences in accuracy

Despite their overall accuracy as a group, participants varied substantially in individual-level accuracy, ranging from perfect to worse than chance. Why could some people report their choice process more accurately than others?

We first examined whether the variation could be explained by several confounding variables, such as participants' choice method (some choice methods might be easier to report than others), quality of model fit, or level of understanding of the self-report questions. These confounders did each significantly predict accuracy—but they did not together explain more than a quarter of its total variance (for weight accuracy, adjusted $R^2$ estimated via linear regression = 3.5% [Study 1A], 21% [Study 1B], 27% [Study 2]; for heuristic error, adjusted $R^2 = 17\%$ [Study 1A], 19% [Study 1B], 2.3% [Study 2]; see SI 3.7). There is thus substantial variation in accuracy that is not explained by these basic confounds.

To further characterize this variation, we collected many individual difference measures, including self-report scales related to introspection (e.g., the self-reflection and insight scale[76]); an attention network test[77]; a measure of cognitive/reasoning ability[78]; experience with the choice domain; and more (see SI 1.3 for full list). To test

whether any measures predicted accuracy, we separately regressed weight and method accuracy on each measure, controlling for the confounding variables described above (see SI 3.8 for details).

After Bonferroni-correcting for multiple comparisons, no measures significantly correlated with any measure of accuracy (see SI 3.8, Tables S7, S8). Even without correcting for multiple comparisons, no measures showed a strong or consistent relationship with awareness. Thus, participants' variation in accuracy is not reducible to other related cognitive constructs, including self-report scales of introspection. This suggests the feeling of insight into one's choice processes is dissociable from a behavioral measure of introspective accuracy.

## Probing the mechanisms underlying accuracy

As described above, there are multiple potential mechanisms underlying participants' ability to accurately report the choice process they had used. One possibility is that participants had direct introspective access to that choice process. But two alternate possibilities are that participants could have inferred their choice process from observing their own choices[17–19,69], or relied on shared, culture-wide background knowledge or lay theories about how people generally make choices in each domain[8]. In other words, rather than relying on privileged first-person awareness of their choice process, participants could instead be examining their choice process as an outsider would, applying general theories or making inferences from observation.

Following past approaches[8,51], to control for these possibilities we recruited an independent sample of "observer" participants in Studies 3A-3B. Each observer was paired with an original, "decider" participant from Study 1A (in Study 3A) or Study 2 (in Study 3B). Instead of making their own choices, each observer was instead shown the decider's choices and asked to report the attribute weights and choice method they believed the decider used. When making these reports, observers

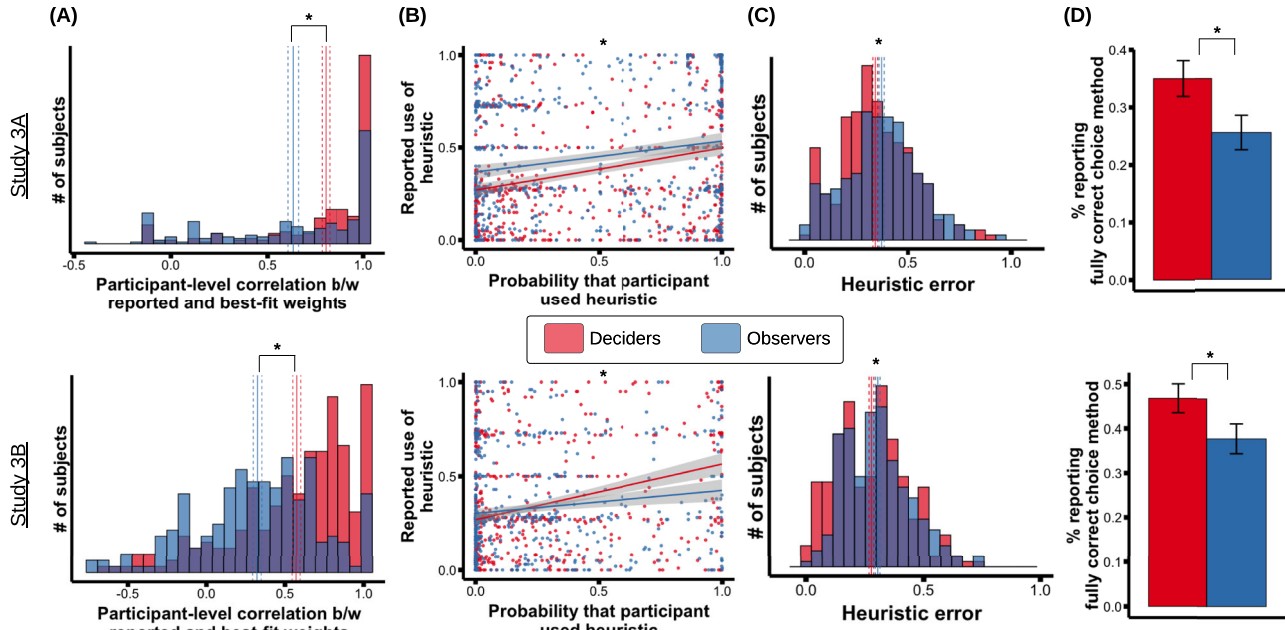

**Fig. 4 | Differences in accuracy between deciders and observers. A** Histogram of participant-level correlations between reported and best-fit attribute weights, as a function of whether the participants were deciders (red) or observers (blue). Solid lines show the mean for each group, with dashed lines showing SEM. Deciders exhibited significantly higher correlations than observers ($p < 0.001$ for both plots). **B** Reported use of each heuristic plotted against the model-estimated probability that the heuristic was used. The least-squares trend line is plotted (with a 95% confidence interval band around it). The interaction between slopes was significant (for top plot, $p = 0.012$; for bottom plot, $p = 0.001$). **C** Histogram of participants' heuristic error, as a function of deciders vs. observers. Solid line shows the mean of each group; the dashed lines show the SEM. Deciders had significantly less heuristic error (for top plot, $p = 0.037$; for bottom plot, $p = 0.025$). **D** Percentage of deciders vs. observers who reported the fully correct choice method; this percentage was higher in deciders than observers (for top plot, $p = 0.031$; for bottom plot, $p = 0.033$). Error bars indicate SEM. In all plots, asterisks indicate a significant effect ($p < 0.05$). For all top plots (Study 3A), there were $N = 237$ deciders and 212 observers; for all bottom plots (Study 3B), there were $N = 235$ deciders and 209 observers.

had access to the same observable information as deciders did, and on average knew the same shared, culture-wide background knowledge about how choices are made. Thus, if deciders were more accurate about their own choice process than observers, this pattern would be consistent with the hypothesis that deciders had direct introspective access to that process.

One issue with this approach is that observers might be less motivated than deciders to pay attention to the task. To counter this possibility, observers were incentivized for accuracy: They were paid a monetary bonus in proportion to how accurately they reported the deciders' choice process (as judged by the computational modeling results). Since deciders were not given incentives for accuracy, if deciders still outperform observers, this result would be strong evidence for deciders' first-person advantage.

For Studies 3A and 3B, we recruited 300 Prolific participants each (nationally representative for age, race, and gender). After applying the same exclusion criteria as before, there were 212 (Study 3A) and 209 (Study 3B) participants for analysis.

We first analyzed weight accuracy. Deciders were substantially more accurate about the attribute weights they had used than observers were (Fig. 4A). We regressed the participant-level correlation between reported and fitted weights on whether the participant was a decider or observer ("participant type"), and found that deciders had significantly higher correlations (Study 3A: difference in mean decider vs. observer correlation = 0.17, participant type $\beta = 0.49$, $t(210) = 5.7$, $p < 0.001$, 95% CI for $\beta = [0.32, 0.66]$; Study 3B: difference in mean correlation = 0.25, $\beta = 0.63$, $t(270) = 7.8$, $p < 0.001$, 95% CI for $\beta = [0.47, 0.79]$) and lower absolute weight error (Study 3A: $\beta = 0.34$, $t(211) = 4.7$, $p < 0.001$, 95% CI for $\beta = [0.20, 0.48]$; Study 3B: $\beta = 0.56$, $t(202) = 7.2$, $p < 0.001$, 95% CI for $\beta = [0.41, 0.71]$).

Another way to analyze these data is to ask what percent of the variance in best-fit weights was explained by deciders' vs. observers'

reported weights. We found that deciders' reports explained 2x as much variance (in Study 3A) and 5x as much variance (in Study 3B) as observers' reports (see SI 4.2 for details). These patterns indicate that deciders were not merely inferring their attribute weights from observing their own choices, or invoking shared culture-wide lay theories, but were additionally relying on some fount of information unavailable to third-party observers—plausibly, direct introspection.

The results were similar for method accuracy. First, we tested whether the self-reported use of each heuristic showed a stronger correlation with the model-estimated probability among deciders compared to observers (Fig. 4B). We regressed the reported use of each heuristic on the model-estimated probabilities, participant type, and their interaction. The interaction was significant (Study 3A: $\beta = 0.13$, $t(1100) = 2.5$, $p = 0.012$, 95% CI for $\beta = [0.025, 0.23]$; Study 3B: $\beta = 0.17$, $t(1040) = 3.2$, $p = 0.001$, 95% CI for $\beta = [0.068, 0.28]$). Moreover, the deciders' heuristic reports again explained 2x (Study 3A) and 5x (Study 3B) more variance in the model-estimated probabilities than observers' reports did (see SI 4.2). Finally, deciders exhibited less heuristic error than observers (Fig. 4C; Study 3A: $\beta = -0.18$, $t(240) = 2.1$, $p = 0.037$, 95% CI for $\beta = [-0.35, -0.011]$; Study 3B: $\beta = -0.2$, $t(280) = -2.2$, $p = 0.025$, 95% CI for $\beta = [-0.37, -0.024]$). These results indicate that deciders could more accurately report which individual heuristics they used compared to observers.

Deciders were also more accurate at reporting which overall combination of heuristics they had used. Their reported models had better Bayes factors (although the effect was only marginal in Study 3B; Study 3A: $\beta = 0.23$, $t(260) = 2.6$, $p = 0.009$, 95% CI for $\beta = [0.06, 0.4]$; Study 3B: $\beta = 0.14$, $t(220) = 1.6$, $p = 0.1$, 95% CI for $\beta = [-0.03, 0.32]$), and they were more likely to report the fully correct choice method (Fig. 4D; Study 3A: $\beta = 0.50$, $z = 2.2$, $p = 0.031$, 95% CI for $\beta = [0.044, 0.96]$; Study 3B: $\beta = 0.48$, $z = 2.1$, $p = 0.033$, 95% CI for $\beta = [0.038, 0.91]$). Together, these findings indicate that deciders were not just inferring

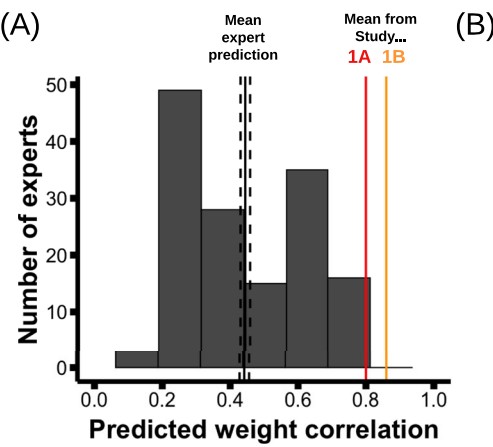
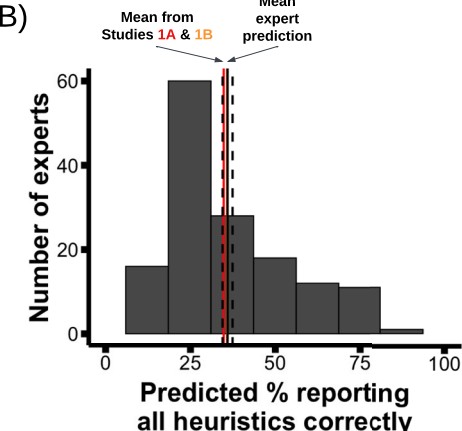

**Fig. 5 | Expert predictions about weight and method accuracy.** Expert predictions of (**A**) the average individual-level correlation between reported and best-fit weights that participants would exhibit in Studies 1A–1B, and (**B**) the percent of participants who would report all three heuristics correctly. In both plots, the black solid line indicates the average expert prediction (with black dashed lines indicating SEM); the red and orange lines indicate the observed values from Studies 1A and 1B, respectively.

their heuristics by observing their own choices, or via cultural lay theories. Instead, they were (at least partly) using some mechanism unavailable to observers—again, plausibly direct introspection.

One concern with Studies 3A-3B is that deciders could have had access to private but non-introspective information that observers did not. For instance, deciders could have idiosyncratic lay theories about their own decision-making, perhaps tailored to their own demographics (e.g., a Manhattanite might infer that they're unusually willing to sacrifice home size for location)—and these idiosyncratic theories could explain how they outperform observers. One prediction of this account is that observers who are more demographically similar to their paired decider—and therefore more likely to have overlapping lay theories about how home/movie choices get made—would exhibit more accuracy. In contrast, we found that the demographic similarity between deciders and observers had no relationship with observers' accuracy (e.g., in both studies, demographic similarity to deciders explained only 1% of the variance in observer accuracy, with average Bayes factors >5 in favor of the null; see SI 4.1 for details). This result suggests that idiosyncratic lay theories, at least demographically influenced ones, do not explain the difference in accuracy between deciders and observers.

### How well can experts predict introspective accuracy?

Finally, we examined how participants' accuracy in these studies compared to the prior expectations of experts in the field. By asking decision scientists to predict how accurate participants will be, we could establish a benchmark against which to compare our results, and test whether participants' accuracy is surprisingly high (as suggested by, e.g., ref. [56])—or instead, whether experts' beliefs about introspective accuracy are already well-calibrated.

In Study 4, we collected predictions from a sample of 170 decision scientists by advertising at the Society for Judgment and Decision Making (SJDM) annual conference and on the SJDM email listserv. We will refer to participants in this sample as "experts". We asked experts to complete the study only if they had a PhD, or were currently in a PhD program. We dropped 23 experts who had heard of our ACP task before, leaving 147 for analysis. The final sample consisted of 54 faculty, 21 postdoctoral scholars, 46 graduate students, and 26 in other roles (e.g., outside of academia; see SI 1.6 for full details about the sample and the recruitment procedure).

Experts in Study 4 were given a brief description of the ACP task from Study 1A-1B, including an example choice trial between homes to rent, a description of how participants in the ACP task reported their attribute weights and choice method, and a description of how we computed weight and method accuracy. When describing how we computed weight accuracy, we focused exclusively on the individual-level correlation between reported and best-fit attribute weights (e.g., the measure in Fig. 2B); when describing how we computed method accuracy, we focused exclusively on the percentage of participants who reported all three heuristics correctly (e.g., the measure in Fig. 3D). We informed experts that the participants were recruited on Prolific, that the sample was nationally representative for age, race, and gender, and that participants were told at the start to pay attention to how they made their choices. Finally, we asked the experts to predict how accurate participants would be: how high the average individual-level weight correlation would be, and what percent of participants would report their use (or lack of use) of all three heuristics correctly (See SI 1.6 for all study materials).

Experts predicted that participants would be substantially less accurate about their attribute weights than they actually were (Fig. 5A). The average individual-level correlation between reported and best-fit weights in Studies 1A-1B was 0.80 and 0.86, respectively. In contrast, the average expert predicted that the average correlation would be 0.44 (95% CI = [0.41-0.47]; this value is significantly lower than 0.80, one-sample t-test, $t(146) = -25$, $p < 0.001$, Cohen's $d = -2.1$, 95% CI for Cohen's $d = [-2.4, -1.9]$). In fact, only 2% of the sample—3 experts out of 147—predicted a correlation of 0.80, and none predicted a higher correlation. Interestingly, senior faculty were the least accurate, predicting an average weight correlation of 0.32 (which was significantly further from the truth than the predictions of more junior experts such as graduate students, $t(17) = 3$, $p = 0.01$, Cohen's $d = -0.94$, 95% CI for Cohen's $d = [-1.95, -0.27]$; see SI 5 for details). The results are similar when comparing experts' predictions to just the participants who used multiple attributes in their decision-making. Finally, though experts' predictions were specifically about the home-rental variant from Studies 1A-1B, their predictions were also lower than the average correlation observed in the movie paradigm of Study 2 (where the average $r$ was 0.57; $t(146) = -9.1$, $p < 0.001$, Cohen's $d = -0.75$, 95% CI for Cohen's $d = [-0.94, -0.60]$), albeit with a less extreme disparity.

Experts' predictions about method accuracy were much more aligned with our experimental results (Fig. 5B). In Studies 1A-1B, the percent of participants accurately reporting all three heuristics was 35% and 36%—and the average expert prediction was 36% [33-39%]. Thus, despite method accuracy never (to our knowledge) having been measured before in multi-attribute choice, experts' predictions about method accuracy were well-calibrated.

Together, these results show that, while the observed levels of method accuracy in our studies were consistent with experts' predictions, the observed levels of weight accuracy were much higher than experts predicted.

## Discussion

How much introspective access do people have to the mental processes used to make value-guided, multi-attribute choices? Despite the ubiquity of these choice contexts[41] and the vigorous empirical debates about awareness of other kinds of mental processes[3], little research has directly, rigorously tested this question. Here we find that people can report key aspects of the choice processes they used with substantial—and sometimes startling—accuracy. Participants reported the decision weights they had placed on each attribute and the overall method they had used to make their choices (i.e., which heuristics they did or did not employ) far more accurately than chance: The average correlation between each participants' best-fitting attribute weights and their self-reported weights ranged across studies from 0.57–0.86, and between one third and one half of participants reported the fully correct choice method (2–3x as many as expected by chance). These are core aspects of how people make many everyday choices—aspects that some prior research has claimed people largely do not know about themselves[8,10,22,79]. Indeed, a sample of decision scientists predicted that people would know substantially less about their own attribute weights than they actually did. Our results show that, though people are far from perfect, they do in fact know much about how they're making choices, at least in the controlled settings and domains tested here.

Moreover, participants knew their own attribute weights and choice method better than informed observers could infer; the correlation between the best-fit weights and deciders' reported weights was (on average) 27-78% higher than the analogous correlations for observers across studies, and deciders were 24−35% more likely than observers to report the correct choice method. These results indicate that people are not simply inferring their choice process from observing their own choices, or relying on shared cultural lay theories about how choices are made. Instead, they have some kind of privileged, first-person knowledge—plausibly acquired via introspective awareness. These findings add to a growing body of literature showing that people may have some introspective access to a surprisingly wide array of mental processes[3,27,30,36,40,75,80,81].

At the same time, while people on average appeared aware of the choice processes they had used, this awareness varied substantially across individuals. Participants' accuracy varied from near perfect to worse than chance. Moreover, this variation was not explained by any individual differences we measured. These results argue against the idea that these choice processes are intrinsically accessible or inaccessible to introspection[8,79], and suggest instead that awareness may vary across people for other reasons (discussed below).

These results have substantial implications across psychology and philosophy. They push back against psychological frameworks that relegate most important choice mechanisms to the unconscious, or that paint decisions as being hugely influenced by factors without awareness of that influence[8,10,22,23,79,82] (at least in the kinds of controlled choice contexts and domains tested here). In contrast, they are more consistent with alternate frameworks that place conscious processing closer to the center of human decision-making[3,46,83], even while participants were still far from perfect at reporting all aspects of their choice process. Finally, there is prominent literature emphasizing people's lack of agency or responsibility over their choices due to the inherent unconsciousness of choice processes (e.g., literatures on nudging or other paternalistic choice interventions[24,25,84]). Our results suggest that people may have more awareness of and—potentially—more ability to directly control their everyday choices than these frameworks suggest, with implications for philosophical accounts of

agency[85,86], clinical psychology[29,32], and interventions to produce behavior change[87].

### Potential mechanisms underlying accuracy

Of course, the present studies do not definitively establish that participants are achieving accuracy about their choice process via direct introspection. For one, although deciders were more accurate than observers, observers still achieved substantial accuracy, demonstrating that some accuracy can be achieved through non-introspective means. A key task for future work will be to test whether deciders' mechanisms for reporting on their choice processes partially overlapped with those of observers, or whether observers were achieving their accuracy in fundamentally different ways.

In addition, the edge that deciders displayed over observers could potentially be explained without appeal to introspection: Deciders could have inferred their choice process from knowledge about themselves not given to the third-party observers (e.g., memories of past choices between homes), or relied on idiosyncratic lay theories about themselves that observers would not share (e.g., "I'm the kind of person who would really hate a small home")[8,56]. On these accounts, participants would still have a first-person advantage, but it would come from mechanisms other than direct introspection.

We believe these alternatives are unlikely to fully explain the results. For one, on these accounts, participants who have more experience with each choice domain (i.e., choosing between homes to rent or movies to watch) would likely exhibit more accuracy about their choice process, due to having more data from which to make inferences or more well-formed lay theories. However, participants' reported experience with each choice domain did not correlate at all with accuracy in any study (see SI 3.8). Moreover, the choices in the ACP task (presented in attribute tables) are not actually that similar to those participants would likely have made in the past (i.e., in past choices between real homes or movies), making it less likely that they would be able to reference memories of or invoke lay theories about past choices when reporting their choice process. Finally, as described in "Results", the demographic similarity between deciders and observers had no relationship with observers' accuracy, further suggesting that idiosyncratic theories are not explaining the difference in accuracy between deciders and observers.

Nonetheless, none of these data definitively rule out deflationary, non-introspective alternatives. To rule out these accounts, future research could take several experimental approaches. One approach would be to introduce a novel intervention that alters participants' choice process in a way that would be difficult to predict a priori, and then test if participants can accurately report how their process changed as a result. For example, subtle visual cues have been shown to influence participants' multi-attribute choice processes[88–91]. If participants shown these cues could accurately report the ways their choice process changed as a result, this pattern would be difficult to explain without introspection. A second approach would be to identify an introspection-related manipulation that improves accuracy without plausibly improving people's inferences or lay theories. For instance, several authors have proposed that introspection into choice processes involves paying attention internally to those processes while they are occurring[27,30,92]. If guiding participants to attend internally this way improves accuracy, this result would be difficult to explain via lay theories or inferential mechanisms, and would further suggest that accuracy is being driven by introspection.

### Individual-level variation in accuracy

Although participants were on average far more accurate than chance, they also varied substantially in their accuracy in ways that were not fully explained by the confounds or individual differences we measured. This finding runs counter to the common assumption that accessibility to introspection is always an intrinsic property of a mental

process[93], and instead suggests that introspective awareness may vary across people in not-yet-understood ways.

What causes this variation? It could just be due to measurement noise or variation in attention or effort invested in the task, in ways that were not captured by our attention and comprehension measures. Future work could test these possibilities by examining what percentage of the variation is preserved across test-retest methodologies.

But there are other explanations for the variation worth investigating. For instance, theories from clinical psychology posit that internal attention is necessary for awareness of mental processes[27,29], and many successful therapeutic interventions are thought to work in part by training internal attention to such processes[30–32,34,35]. Thus, one plausible explanation of the variation in accuracy is that participants varied in the extent to which they were paying attention internally while their choice process was occurring, perhaps due to variation in internal attentional capacity or motivation to attend internally. A promising direction for future research could be to test whether measures of internal attentional capacity/motivation predict variation in accuracy on the ACP task, or whether improving or guiding internal attention can lead to improvements in accuracy[27].

## Limitations of the present studies

The present studies have several limitations. One key limitation is the generalizability and ecological validity of our task. By presenting constrained, artificial choices (as most past research on multi-attribute choice has done[67]), we were able to precisely model participants' choice processes and quantify their accuracy. However, most real-world choices are not presented in attribute tables[41], and it is plausible that, when attributes are presented less explicitly, their influence is less accessible to awareness. Moreover, participants' choice processes in these artificial contexts may have a limited correspondence to the processes people would use in real-life versions of these choices—e.g., the attribute weights employed in these contexts may not be representative of the weights people would actually place on analogous attributes in real life. A key future direction to address both these concerns will be to probe accuracy in more realistic choice formats. Future research should also test whether participants' accuracy generalizes to other domains of value-guided choice. For instance, in social choices, there may be additional obstacles to accuracy: People may be motivated to remain ignorant of the social biases influencing their choices, or may not want to reveal their knowledge for desirability reasons[94,95].

A related concern is that our results speak only to awareness of value-guided, multi-attribute choice processes; other kinds of choice processes may be differently accessible to introspection[3]. A particular concern is that the choice contexts studied in our paradigm may have stacked the deck in favor of awareness. For instance, by focusing on repeated, low-stakes choices with many attributes, we may have pushed people towards simpler strategies (such as the "take the best" heuristic) that are more likely to be consciously accessible. This concern is partly alleviated by the fact that the participants who used more complex strategies (e.g., considering many attributes at once) still exhibited substantial awareness of their choice process in our experiments. Moreover, the same features of our experiments could plausibly push people away from awareness: People may pay less attention to low-stakes choices[96], and the fact that participants repeat the choices many times could push them towards habitual "System 1"-type decision-making[97]. Ultimately, the field lacks a consensus theory of when people will be aware or not of their choice processes; by rigorously mapping out awareness across different choice contexts, we get closer to building that theory, and the methods developed here can be further leveraged towards this goal.

An additional concern is that our studies rely on computational model-fitting to ascertain the ground truth of key elements of participants' choice processes. While computational modeling is a reliable method for measuring choice processes[41,43,45,98], our models obviously cannot capture all the myriad ways people might make multi-attribute choices. For instance, our models assume that participants were using the same weights and choice method consistently throughout the task, and that there were no nonlinear interactions between the attributes. While these assumptions make the modeling tractable, they surely miss some elements of participants' actual choice process[65] (though see SI 3.2 for supplementary analyses indicating that these assumptions were not strongly violated). In addition, computational modeling has been critiqued for relying on black-box prediction rather than testing qualitative, falsifiable patterns in behavioral data[99]. To the extent that our models fail to capture participants' true choice process, we are adding noise into the analysis and likely underestimating participants' accuracy; thus, we consider these accuracy estimates to be lower bounds on participants' true accuracy.

A fourth limitation is that, in our studies, participants report their choice process retrospectively, after their choices have been made. We believe it likely that participants had introspective access to their choice process as it was unfolding, enabling them to report it after. But our experimental design can only speak to participants' ability to report the choice processes they used in the past, not the choice processes unfolding in the present (or the processes they expect to use in future choices). A useful future direction could be to test for introspective awareness of present-moment choice processes with think-aloud paradigms[47,100].

Finally, a limitation of this work is that it focuses on two aspects of value-guided, multi-attribute choice processes: attribute weights and the use of heuristics. But, of course, there are many other aspects of these choice processes which people may have less introspective access to. For instance, it seems less likely that participants would have introspective access to lower-level implementational details of their choice process (e.g., whether they are using a ballistic or leaky-competing accumulator[101]). The important research question is not whether people have access to all aspects of their choice process, but rather which aspects of their choice process (or which choice processes) they have access to, and in what settings. Our results provide strong evidence that one ubiquitous kind of choice process is substantially (though imperfectly) accessible to awareness; a task for future research is to further characterize which processes are more or less introspectively accessible.

## Methods
### Multi-attribute choice processes

The cognitive processes underlying people's multi-attribute choices have several common elements (Fig. 6). People typically represent how they will weigh the attributes relative to each other (e.g., a weight of 0.75 on home size versus 0.20 on kitchen quality); represent where each option falls on each attribute (e.g., Home A is very large but has a poor kitchen); and combine this information to compute an overall value for each option[41,42,74]. Each element of this process can take a variety of forms. For example, in one common form (typically considered the "classically rational" model of multi-attribute choice), people assign graded weights to each attribute, capturing the relative importance of each for their choices; represent the graded values of each attribute for each option, scaled to some standard range (in our case, between 0 and 1); and then combine this information linearly to compute overall values for each option[41,42,44,63]. This model is also sometimes called the "weighted additive decision" rule[42,63]; see SI 2.1 for mathematical details of all choice models.

People, however, often employ heuristics which simplify their choice process in various ways. Here, we focus on three core simplifying heuristics that are widespread in two-alternative multi-attribute choice and that can be measured via computational modeling (Table 1)[41,44,62,63,73].

## Elements of multi-attribute choice processes

| Element | Example | |
|---------|---------|---|
| Represent how you will weight the attributes (between -1 and 1): | $w_{size} = 0.4$<br>$w_{kitchen} = 0.25$<br>... | |
| Represent each option's attribute values (scaled to between 0-1): | $size_{HomeA} = scale(1800\ sq.\ ft.) = 1$<br>$kitchen_{HomeA} = scale("Moderate") = 0.5$<br>... | $size_{HomeB} = scale(600\ sq.\ ft.) = 0$<br>$kitchen_{HomeA} = scale("Very\ Good") = 1$<br>... |
| Compute overall values of each option: | $value_{HomeA} = w_{size} * size_{HomeA} +$<br>$w_{kitchen} * kitchen_{HomeA} + ...$ | $value_{HomeB} = w_{size} * size_{HomeB} +$<br>$w_{kitchen} * kitchen_{HomeB} + ...$ |

**Fig. 6 | Common elements of value-guided multi-attribute choice processes.** People typically form decision weights for each attribute, representing how much they weigh that attribute in their choice relative to the other attributes. They also represent each option's attribute values, scaling them to make them comparable. Finally, they combine these weights and attribute values to compute overall values for each option. When the weights and attribute values are graded, this process is the "classically rational" model of multi-attribute choice (or the "weighted additive decision" rule). Common choice heuristics simplify this process by simplifying the representations of the weights or attribute values.

**Table 1 | Three heuristics people often use to simplify two-alternative multi-attribute choice, and how we implement them computationally**

| Heuristic name | Heuristic description | Computational changes to choice process | Notes |
|----------------|-----------------------|------------------------------------------|-------|
| Single attribute (or "take the best", "lexicographic") | Instead of integrating multiple attributes together, choose based on a single decisive attribute[62,102,103]. | The attribute weights become 1 (or -1) for one attribute and 0 for all others. | In our task, this heuristic is equivalent to Elimination by Aspects[104]. |
| Binary weights (or "equal weights") | Instead of assigning graded attribute weights, weight all included attributes equally[73]. | The attribute weights become binary: either 1 (or −1) or 0. | Here, we adapt this heuristic to allow some attributes to not be considered (i.e., have a weight of zero). |
| Binary attribute values (or "weighted pros") | Instead of representing graded attribute values, represent which option is greater on each attribute[44,105]. | The attribute values become binary: either 1 (for the option with the greater value on that attribute) or 0 (for the other option). | People using this heuristic do not compute stand-alone values for each choice option; instead, they only consider the ways that one option is better or worse than the other. This is similar to the process in decision by sampling[64]. |

As a first heuristic, instead of integrating multiple attributes together, people sometimes make choices based on a single attribute[62,102,103]. This heuristic is sometimes called the "lexicographic"[103] or "take the best" heuristic[62,102]. We implement this heuristic computationally by constraining the attribute weights such that exactly one attribute receives a weight of 1 (or −1) and all others a weight of zero (In the two-alternative forced choice setting of our task, this heuristic is equivalent to Elimination by Aspects[104]).

Second, when people integrate attributes together, instead of assigning graded weights to those attributes, sometimes people simply represent whether each attribute is included or not in their choice process and weight all the included ones equally[73]. We implement this computationally by constraining attribute weights to be binary: either 1 (or −1), or 0. This heuristic is sometimes called the "equal weights" heuristic (although our implementation differs slightly from the original heuristic by allowing some attributes to be excluded from consideration). Intuitively, people using the binary weights heuristic are simply representing which options are "in" consideration and which are "out", and then not differentiating further between the "in" options (i.e., weighing them all equally).

The third heuristic concerns people's representations of the attribute values: the value of each option on each attribute (e.g., the size of Home A). In our implementation of the classically rational model, people represent the actual attribute values, scaled within the attribute's range (e.g., if Home A is 2000 sq. ft. and the range is from 1000 to 3000 sq. ft., then Home A's represented value for size might be 0.5). But a simpler, alternative approach is to just represent which option is better on each attribute[64,105]. In this case, the attribute values can be modeled as binary: 1 for the option with the greater value, and 0 for the other

option. Concretely, if Home A is 2000 sq. ft. and Home B is 1600 sq. ft., Home A would receive a size value of 1 and Home B of 0. This heuristic thus implements a core insight from, e.g., decision-by-sampling models: People sometimes do not compute standalone values of decision options, and instead just compare options to their immediate alternatives[64].

As stated above, these heuristics are not meant to capture all of the cognitive processes underlying multi-attribute choice. Rather, they are well-established elements of people's choice processes that we can measure precisely via computational modeling[41]. The ACP task leverages these models to precisely measure key elements of participants' multi-attribute choice processes and quantify participants' accuracy about those processes.

### Details of the ACP task

**Procedure.** In Part 1, participants first received detailed instructions about the choices they would be making between hypothetical homes to rent (Study 1) or real movie trailers to watch (Study 2; see SI 1.1–1.2 for full instructions). They then completed 10 practice choice trials, followed by 100 real choice trials. (Simulations revealed that the large number of choice trials was necessary to accurately measure participants' individual attribute weights and choice method; see SI 2.3 for simulation details.)

On each choice trial, participants were shown two options described in an attribute table with nine attributes (e.g., Fig. 1A), and were asked to choose one. The full set of attributes is in SI 1.1. For all choice trials, in order to encourage participants to read the options carefully and not just click quickly through, we required participants to wait 7 s before making their choice. Participants were instructed at the start of the study to pay attention to how they're making choices, but

were not told in advance that they will be asked to report upon their choice process.

In Study 1A, we pre-selected a fixed set of choice trials, while randomly shuffling the order of those trials for each participant. In Study 1B, we randomly generated the set of choice trials anew for each participant. For both studies, on all choice trials, we ensured that the two homes differed substantially from each other on each attribute (full details in SI 1.1). In Study 2, we pre-selected a fixed set of choice trials, drawing from the MovieLens database which aggregated user ratings of real movies along various dimensions (see SI 1.1 for details refs. [70–72]). For all trials, we randomized the order of the attributes in the attribute table and the order of the two home options.

In Part 2, participants were first asked to describe in their own words how they made choices in Part 1 (we do not analyze this free response in this manuscript; see SI). Then, they completed two types of guided self-reports: reports about the heuristics they used and the attribute weights they used (we randomized whether the heuristic or weight self-report section was presented first).

In the heuristic self-report section, for each heuristic in Table 1, we first presented a detailed, intuitive description of the heuristic (the order of the three heuristics was randomized). For each heuristic, we described two "strategies" that a person could use to make choices in Part 1 of the task. In each case, "Strategy A" was using the heuristic, and "Strategy B" was not using the heuristic. For example, for the single attribute heuristic, we told participants that a person using Strategy A made all their choices based on a single, all-important feature, while a person using Strategy B considered multiple features of the homes when making their choices. After describing Strategy A vs. B, we asked two comprehension check questions (for the exact text of the descriptions and comprehension checks, see SI 1.2).

Then, critically, we asked participants to rate the extent to which they believed the heuristic accurately described how they made their choices, on a continuous scale from "Strategy A much better describes how I made my choices" to "Strategy B much better describes how I made my choices". Finally, as an exploratory question, we asked participants to rate whether Strategy A or B better described how they believed one *should* make their choices in Part 1; we do not analyze this question in this manuscript.

Measuring people's beliefs about their attribute weights is complicated by the fact that participants might represent different formats of attribute weights depending on which heuristics they are using. A participant using the classically rational process (i.e., none of the heuristics in Table 1) will represent graded weights (i.e., from 0 to 1); a person using the binary weights heuristic will represent binary weights (i.e., each attribute has a weight of 1 or 0); and a person using the single attribute heuristic will represent weights that equal 1 for one attribute and 0 for the others.

To elicit participants' self-reported attribute weights without assuming which heuristic(s) they were using, we asked them to report their weights in all formats: graded, binary, and single-attribute (the order of the three types was randomized). To elicit graded attribute weights, we asked participants, for each attribute, to report the extent to which they had cared about that attribute when making their choices (on a 100-point scale from "Didn't care about it at all" to "It completely determined my choices", with 9 labels to mark spots on the scale; see SI 1.2 for exact scale). To elicit binary attribute weights, we presented a list of the attributes with check boxes next to them, and asked participants to check *which* attributes they had cared about when making their decisions. To elicit single attribute weights, we presented the list of attributes with radio buttons next to them, and asked participants to select which single attribute they had *most* used to make their decisions.

Note that all of these questions ask about the *magnitude* of participants' attribute weights, but not their *direction* (or sign; i.e., whether each attribute contributed positively or negatively to their

evaluation of a home). Thus, we also asked participants to report the direction of their attribute weights. We implemented these direction questions differently in Studies 1A vs. later studies. In Study 1A, we showed participants the range of possible values each attribute could have taken on, and asked them to report which attribute they would have most preferred and which they would have least preferred (all else being equal). However, participants reported finding these questions confusing; hence, in all later studies, we made these questions more intuitive. For each adjacent pair of possible values each attribute could have taken on (e.g., if the attribute scale was {Very Bad, Bad, Moderate, Good, Very Good}, one adjacent pair would be {Very Bad, Bad}), we asked participants to imagine two options, one with each attribute value (and equal on all other attributes)—and then to report which of the two options they would have preferred (see SI 1.2 for details).

Part 2 also contained several additional quality control questions; see SI 1.3 for details. The task was coded in Javascript using jsPsych v6.3[106]. The code for the attention network task in the individual difference battery was created by ref. [107].

**Model-fitting.** The three heuristics in Table 1 combine to form six potential overall models of people's choice process[41]: (1) none of the three heuristics (the "classically rational" model); (2) just the binary attributes heuristic; (3) just the binary weights heuristic; (4) both the binary weights and binary attributes heuristic; (5) just the single attribute heuristic; and (6) the single attribute and binary attributes heuristic. The binary weights and single attribute heuristics cannot be combined, because the latter is a special case of the former.

For all models, we assumed participants convert option values into final choices via a softmax function with a free inverse temperature parameter (see SI 2.1 for details). Thus, the parameters for each model are an inverse temperature and nine attribute weights, whose constraints differ based on the model: for models #1–2, the weights range continuously from −1 to 1; for models #3–4, the weights take on discrete values {−1, 0, 1}; and for models #5–6, the weights are constrained such that exactly one weight takes on a discrete value {−1, 1}, while the others equal zero. We assumed uniform prior weight parameters and a Gamma(4, 1) prior for the inverse temperature parameter.

All model-fitting was done in R version 4.3.1. We fit the models to participants' choices with Stan[108] (version 2.32.6), which uses contemporary Bayesian methods to estimate the posterior distribution of parameters given the participants' choices. Using the Stan results, we computed the maximum a posteriori weights for each model for each participant. For each model, we then estimated the marginal likelihood (i.e., the probability of the participant's choices given the model, marginalizing over the model's free parameters) using bridge sampling[109]. Normalizing over the marginal likelihoods gave us the posterior probability of each model given the participant's choices. This method of model comparison implements a Bayesian Occam's razor: More flexible models, such as the classically rational model, will automatically be penalized for having more flexible free parameters[110,111]. See SI 2.1 for details about our modeling procedure.

Note that each choice method yields a different set of best-fit attribute weights—making the two elements of choice processes that we measure (weights and choice method) interdependent. To account for this and maximize independence between the two elements, we computed a set of "model-averaged" best-fit attribute weights, averaging the weights across the six choice methods weighted by the posterior probability of each method (a well-established technique[112]). We then did the same procedure to the three types of self-reported attribute weights—graded, binary, and single-attribute—to compute model-averaged self-reported weights, which could then be compared to the model-averaged best-fit weights. All analyses of attribute weights in the main text use these averaged weights. Importantly, none

of our results hinge on this model averaging approach; they remain qualitatively similar using alternative approaches to analyzing the attribute weights (see SI 3.9).

We then used the Stan output to compute several more values. We extracted the posterior probability that participants used each heuristic separately, via Bayesian model family comparison[113]: We partitioned the choice models into "families" that either used or didn't use each heuristic, and then computed the posterior probability of each family given each participant's choices (see SI 2.2 for details). This procedure allowed us to analyze each heuristic separately. Finally, we performed several procedures to assess the acceptability of the Stan model fits (such as split-half reliability estimates); see SI 3.2 for details.

Using these model-fitting results, as our primary measure of weight accuracy, for each participant we computed the Pearson correlation between their nine model-averaged best-fit weights and nine model-averaged self-reported weights (one for each attribute). Note that the best-fit and self-reported weights are both continuous measures ranging from −1 to 1, with neither constrained to sum to a constant, making the two comparable. We also computed the absolute weight error by averaging the absolute difference between the best-fit weight and self-reported weight across the nine attributes. For details of how we computed the information-gain measure of weight accuracy, see SI 3.9.

For each participant, we also computed several measures of method accuracy. We computed "heuristic error" as the absolute difference between the posterior probability and reported use of each heuristic (scaled to 0-1), averaged across the three heuristics. To compute the "Bayes factor of the reported model", we first identified the overall choice model that the participant reported as most likely to characterize their choices by dichotomizing each of their heuristic reported uses into either "used heuristic" (extent >50%) or "did not use heuristic" (extent <50%), which leads to one of the six models described above. Then, we computed the ratio of the posterior probability of the reported model to the posterior probability of the best-fitting model. We determined the chance levels of each of these measures via bootstrapping simulation; see SI 2.5 for details.

**Simulations.** To test whether our analysis can correctly identify the choice model (i.e., heuristics) and attribute weights producing choice data, we simulated data from decision-makers using a variety of heuristics and weights. We then ran the model-fitting procedure on each simulated decision-maker, and compared the results with the models/weights we knew produced their choices. This procedure identified the correct heuristics for over 80-90% of simulated decision-makers, and the fitted attribute weights showed near-perfect correlation with the weights used to simulate the data. This result demonstrates that our analysis method can, in principle, accurately identify the heuristics and weights underlying choices in the ACP task. For details of the simulation procedure and results, see SI 2.3.

**Analyses.** Methods for computing inferential statistics are noted in each analysis in the main text above. For all analyses involving multiple data points per participant, we estimated mixed effect regressions, beginning with a maximal random effects structure and then removing effects that captured no variance and prevented convergence (see SI 2.4 for exact procedure). All reported regression coefficients are standardized, and all tests are two-sided. When comparing the deciders and observers within a regression, we included a random effect for which decider the observer had been paired with, thus incorporating the pairing between deciders and observers in the inference (akin to running a paired t-test). Any continuous measures (like the heuristic errors and Bayes factors) were compared to chance with a one-sample t-test; any binary measures (like whether participants reported the correct overall choice model) were compared to chance with a chi-squared proportion test. Since all statistical tests examine different questions, we do not correct for multiple comparisons (except as noted in "Individual differences in accuracy", when we regress accuracy on numerous individual difference measures; see SI 3.8 for details). All analyses were conducted in R 4.3.1, with mixed-effect models estimated via the package *lme4*[114], version 1.1-34 (and corresponding *p* values computed via the package *lmerTest*[115], version 3.1-3).

**Modifications to the ACP task for the observer version.** In the observer version of the ACP task, every observer participant was randomly matched with a "decider" participant from a prior study. The instructions were similar to the decider version, except the observers were told that, instead of making their own choices, they would observe the choices made by a past participant, and that they would earn bonus money in proportion to how accurately they could report how the participant made those choices (as judged by our computational modeling).

In Part 1, on each choice trial, the observers were shown the two options given to their matched decider. After 7 s they were told which option the decider chose, and to proceed to the next trial they had to select the corresponding button that the decider selected (so that the observers would mimic the deciders' actions). Finally, in Part 2, observers completed identical questions as the deciders, except, instead of reporting how they had made their choices, they reported how they believed the decider made their choices.

### Study details

We recruited 300 Prolific participants for each of Studies 1–3. Studies 1A, 1B, 2, and 3A consisted of two sessions, run <1 week apart; the first session was the ACP task, and the second was a battery of individual difference measures (described in SI, Table S1). Study 3B omitted the second session. All participants gave informed consent and were paid $12/h; all study procedures were approved by the Yale Institutional Review Board (protocol #2000022385) and Princeton Institutional Review Board (protocol #14791). As noted above, all Prolific studies were nationally representative for (self-reported) age, race, and gender. (Note that our individual difference analyses described above did not find any consistent differences across any demographic variables.)

Across Studies 1–3, we excluded participants who did not complete all choice trials; who spent an average of less than 2 s per instruction screen; who got fewer than 4/6 of the comprehension check questions correct; who reported incoherent attribute weight directions (see SI 1.2 for details); who self-reported having paid <50% attention to the study; or who tabbed away from the study more than 20 times. In Studies 1–2, we also excluded participants who chose the right or left option on more than 80% of choice trials; in Study 3, we instead excluded participants who selected the correct button (i.e., the button highlighted as the original decider's choice) on fewer than 95% of choice trials.

After exclusion, 237 participants remained for analysis in Study 1a (53% female, 46% male, 1% other; mean age = 46 ± 16 years); 251 in Study 1b (53% female, 43% male, 4% other; mean age = 39 ± 14 years); 235 in Study 2 (50% female, 45% male, 5% other; mean age = 43 ± 15 years); 212 in Study 3 A (50% female, 48% male, 2% other, mean age = 46 ± 16 years); and 209 in Study 3B (58% female, 40% male, 2% other, mean age = 46 ± 16 years). Note that 22 of these participants did not complete the second session in Study 1a, 45 in Study 1b, 15 in Study 2, and 34 in Study 3A (there was only one session for Study 3B); we include these participants in our main analyses, and only exclude them from the individual-difference analyses requiring data from the second session.

For Study 4, the expert prediction study, we recruited 170 participants from the SJDM annual conference and email listserv (see SI 1.6 for details). We excluded participants who reported having heard of our ACP task before, leaving 147 for analysis (42% female, 55% male, 3% other; mean age = 38 ± 12 years). All participants gave informed

consent, and this study was approved by the Princeton IRB (protocol #17213). Participants in Study 4 were not compensated.

## Reporting summary

Further information on research design is available in the Nature Portfolio Reporting Summary linked to this article.

## Data availability

All raw data generated in this study have been deposited in the OSF database accessible here: https://doi.org/10.17605/OSF.IO/TMQJU.

## Code availability

All code used to run the experiments and analyze data in this study have been deposited in the OSF database accessible here: https://doi.org/10.17605/OSF.IO/TMQJU.

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

## Acknowledgements

We thank Arshiya Aggarwal for her invaluable assistance with programming the experiments, Sudeep Bhatia for his advice on models of multi-attribute choice, and all members of the Crockett Lab and the Clinical Affective Neuroscience Lab for their feedback on this work. NIH Kirschstein-NRSA F32MH131253 and M.J.C.'s discretionary funds from Yale University and Princeton University supported A.M. John Templeton Foundation Grants #61495 and 62816 also supported A.M. and M.J.C. NIH grants R01AA029137, R01DA042911, R01DA043690, and R34AT009887 supported H.K. Data collection was supported by M.J.C.'s discretionary funds from Yale University and Princeton University.

## Author contributions

A.M., R.W.C., H.K., and M.J.C. all contributed to conceptualization, methodology, and writing (review/editing). A.M. was responsible for running the experiments, analyzing the data, and writing the original draft. M.J.C. and H.K. supervised the project, and M.J.C. provided funding.

## Competing interests

The authors declare no competing interests.
