## [Transparent Peer Review file · Nature Communications]

Introspective access to value-based multi-attribute choice processes

Corresponding Author: Dr Adam Morris

Version 0:

Reviewer comments:

Reviewer #2

(Remarks to the Author)

The authors have comprehensively tackled the previous round of comments, and I have no further concerns. I particularly appreciated the more nuanced discussion section.

One thought occurred to me while re-reading the paper that the authors may wish to consider when finalising their files for publication. There was a notable omission of work on metacognition of (eg confidence in) value-based choice - see a couple of classic references copied below, and a significant body of work that has built on this. This work is I think consistent with the authors' findings in showing that people are reliably aware of the values contributing to their choices, in the form of being able to recognise (with low confidence) when their choice does not align to their stated values. While the investigation of confidence in 2AFC decisions is not the same as introspection of multi-attribute choice, this would seem an important point of contact with the metacognition literature (and potentially a source of hypotheses as to the brain systems underpinning introspection in the authors' novel task).

De Martino, B., Fleming, S. M., Garrett, N., & Dolan, R. J. (2013). Confidence in value-based choice. *Nature Neuroscience*, 16(1), 105-110.

Lebreton, M., Abitbol, R., Daunizeau, J., & Pessiglione, M. (2015). Automatic integration of confidence in the brain valuation signal. *Nature Neuroscience*, 18(8), 1159-1167.

Reviewer #4

(Remarks to the Author)

I was asked to step in for the Reviewers 1 & 3 from the previous round of review and to evaluate how well the authors' revision addresses the issues that were raised. Overall, my view is that the authors have mostly, but not completely, addressed the reviewers' concerns. I think there is still a question of whether participants were aware of the relative weighting of non-zero attribute weights.

In the previous round, the reviewers questioned whether the authors were using a straw man argument. I think that the authors have convincingly argued that they are not, that there are prominent articles claiming that people are not able to assess their decision strategies. This is somewhat undermined by other referenced articles that do report positive correlations between reported weights and actual weights (Table S11). Clearly there are arguments on both sides of the debate. The current article adds to that debate.

Another prominent issue was the ecological validity of the tasks. These tasks are similar to many multi attribute decisions in the Judgment and Decision Making & Marketing literatures. There is clearly value in studying these kinds of tasks, if only to address those literatures. It does seem worth noting that the amount of introspection could be very different in other more natural settings. Presenting information in tables allows decision makers to easily screen out attributes that they want to ignore. But when attributes are not nicely separated and organized it may be difficult to ignore those attributes. For example, when purchasing a house, ugly colored walls may make you hate the house, even knowing that you could easily repaint

them and so should ignore that attribute. It is simply not possible to avoid looking at the walls when walking through a house. The current study cannot address examples like this one.

This gets me to the lingering concern that I mentioned in my summary paragraph. With so many attributes, it is quite likely, and consistent with Figure 2A, that people would ignore several attributes, i.e., assign them weights of zero. That would be a very conscious decision to not look at several attributes, in order to simplify these choices. The question is whether this accounts for a substantial amount of the correlation between fitted and reported weights. There are a couple of ways the authors could address this. First, they could use their "information gain" metric but exclude reported weights of 0 when computing the metric. Second, they could examine how often the reported rankings of the top two or three attributes aligns with the fitted rankings of those same attributes. This would focus the analysis on what matters, the weighting of the most important attributes, and not on the irrelevant attributes.

Reviewer #5

(Remarks to the Author)

I was given this manuscript, with its peer-review history, to evaluate the quality of the rebuttal, notably regarding the issues and concerns raised by Reviewers 1 and 3.

In my view, the main reservations of both reviewers concerned two main aspects: 1) the initial framing of the study, which they viewed as an exaggerated refutation of a strawman argument (namely "researchers believe that individuals have no access to their decision criteria"); and 2) the ecological validity of the task, which, by systematically varying many attributes of options over hundreds of trials with limited stakes, could lack generalizability and might incidentally improve introspective accuracy.

I fully agree with the rationale of these initial reservations, yet consider (like reviewer 3) that the manuscript was nonetheless based on a solid methodology (task design, replications, solid sample size, advanced/innovative analytical approach, etc.) and contains material, which, even under a more reasonable framing, could advance the debate on human introspective ability about their decision motives and be of (methodological) interest to a relatively large crowd of researchers.

In their revision, I have the feeling that the authors have seriously amended their manuscript to address most concerns, and reframed their contribution/angle in a much more reasonable and productive direction. I also think some reservations raised by Reviewer 1 were overly critical and required the present work to raise to uncommon standards in behavioral sciences: sure, this work is not perfect and some methodological compromises had to be devised, and sure this work cannot settle the debate once and for all; but, again, I think in its revised form, it constitutes a reasonable, more nuanced, and solid contribution to the scientific question at hand.

Here are a couple of additional (minor) thoughts:

1) I have the feeling that the results are framed in a way that suggest that we evaluate the "true" weights – i.e. motives/preferences- of participants. I have two reservations about this: first, the task is barely value-based (the stakes are pretty low in the context of the hypothetical study 1 and the incentive study 2: watch a movie trailer), so it should be made explicit that those weights might very-well be quite idiosyncratic to the task. Second, and in the same line, there might very well be an interaction between the task design (i.e. the distributions from which are drawn the attribute values) and participants actual valuation of the attributes. For instance, there are some models in multi-attribute choice, that weight attributes as a function of their relative variance in the choice set (because it becomes the main differentiation characteristics). This possibility of context-dependency / idiosyncrasy of the weights should be discussed.

2) Related to this, I would expect to see the distribution of weights splitted per attributes (could be in the SOM) to see how variable/consistent are participants relative weighting of the different attributes. Would there be a way to improve the Figure 2A to include this ? (also because the statistical model used in the Main text do use random intercepts and slopes per individual and attribute).

3) An important interpretational dimension, also, is that introspection is made retrospectively on choices. This means that this manuscript does not investigate participants general introspective accuracy about their preferences/values and its link to choices (e.g. in the future), but a more limited version of introspection accuracy, which tries to rationalize, ex-post, a set of choices. I think the authors should pay more attention to this dimension, to nuance their angle and findings. E.g. in the abstract, a liberal use of the "general" present tense (e.g. "How much awareness do people have about how they make such choices?") could be replaced with a past tense (e.g. "How much awareness do people have about how choices they have made?"). This type of small modification could/should be generalized throughout the abstract and manuscript, and the subtlety briefly discussed in the discussion.

Version 1:

Reviewer comments:

Reviewer #4

(Remarks to the Author)

The authors have done a great job addressing all the concerns. I'm happy to endorse this paper now.

Reviewer #5

(Remarks to the Author)

Again, the authors engaged in a serious and constructive revision process. In my view, most concerns are satisfactorily addressed, and the addition of the new experiment increases the general interest of the paper.

I now recommend the manuscript for publication, thank the authors for their constructive approach to this revision process, and congratulate them on a nice paper.

SUMMARY OF REVISIONS

We are grateful for the Reviewers' feedback. To address the concerns, we have revised the manuscript in several key ways. All major changes are highlighted in blue font in the manuscript file.

1. In response to the concern that the observed weight accuracy could be an artifact of including zero-weight attributes, we re-computed participants' weight accuracy, dropping attributes which participants reported as having zero (or near-zero) weight. We found that, even when only including attributes with nonzero weight, participants were still very accurate about their own attribute weights (nearly as accurate as reported previously; e.g., in Study 1A, the average participant-level correlation between reported and fitted weights dropped from 0.80 to 0.75). We also took Reviewer 4's other suggestion and recomputed weight accuracy with only each participant's top 3 most important reported attributes. Again, we found similarly high levels of weight accuracy for these attributes. These new analyses are reported on page 6 of the revised manuscript.
2. As Reviewer 5 suggested, we have added graphs depicting the variability of weights across each attribute. To keep the main text focused on our central points, we have included these graphs in the Supplement; we reference them in the main text on page 5.
3. In our revised General Discussion (p. 17-18), we have further emphasized the limitations of our task noted by Reviewers 4 and 5 – specifically, around ecological validity, the context-specific nature of the choice processes invoked in our paradigm, and the fact that we assess introspective accuracy retrospectively about past choices.
4. Finally, to emphasize the theoretical advance this work provides, we included a new study. This new study demonstrates that our findings are counterintuitive for decision scientists. Specifically, we asked 147 decision scientists to predict the results of Study 1A-B. We found that these experts significantly underestimated key aspects of choice process awareness, offering support for our claim in the Introduction that many decision scientists believe choice processes operate below awareness. These data are reported below in response to Reviewer 4, and as Study 4 in the revised manuscript.

We include point-by-point responses to each Reviewer comment below.

REVIEWER COMMENTS

Reviewer #2 (Remarks to the Author):

The authors have comprehensively tackled the previous round of comments, and I have no further concerns. I particularly appreciated the more nuanced discussion section.

One thought occurred to me while re-reading the paper that the authors may wish to consider when finalising their files for publication. There was a notable omission of work on metacognition of (eg confidence in) value-based choice - see a couple of classic references copied below, and a significant body of work that has built on this. This work is I think consistent with the authors' findings in showing that people are reliably aware of the values contributing to their choices, in the form of being able to recognise (with low confidence) when their choice does not align to their stated values. While the investigation of confidence in 2AFC decisions is not the same as introspection of multi-attribute choice, this would seem an important point of contact with the metacognition literature (and potentially a source of hypotheses as to the brain systems underpinning introspection in the authors' novel task).

De Martino, B., Fleming, S. M., Garrett, N., & Dolan, R. J. (2013). Confidence in value-based choice. Nature Neuroscience, 16(1), 105-110.

Lebreton, M., Abitbol, R., Daunizeau, J., & Pessiglione, M. (2015). Automatic integration of confidence in the brain valuation signal. Nature Neuroscience, 18(8), 1159-1167.

We thank the Reviewer for their positive feedback on our revised manuscript, and for the pointer to the relevant literature on metacognition of value-based choice. We now incorporate and reference this literature in the introduction (p. 1).

Reviewer #4 (Remarks to the Author):

I was asked to step in for the Reviewers 1 & 3 from the previous round of review and to evaluate how well the authors' revision addresses the issues that were raised. Overall, my view is that the authors have mostly, but not completely, addressed the reviewers' concerns. I think there is still a question of whether participants were aware of the relative weighting of non-zero attribute weights.

We thank the Reviewer for stepping in to complete the review, and for their feedback below.

In the previous round, the reviewers questioned whether the authors were using a straw man argument. I think that the authors have convincingly argued that they are not, that there are prominent articles claiming that people are not able to assess their decision strategies. This is somewhat undermined by other referenced articles that do report positive correlations between reported weights and actual weights (Table S11). Clearly there are arguments on both sides of the debate. The current article adds to that debate.

We agree that there are arguments and existing empirical results on both sides of this debate, and we are glad that the Reviewer found our characterization in the revised manuscript to be more balanced and fair. In the interim between submitting the prior revision and getting this current round of reviews, we put additional thought into the concern that we were mischaracterizing the existing beliefs of the field. We decided to address the concern empirically. We recruited a sample of decision scientists (from the "Society of Judgment and Decision Making" conference and listserv), described our task from Studies 1A-1B, and asked them to predict how accurately people in the task would report their attribute weights and choice method. We found that, while decision scientists indeed had a wide range of predictions, they

substantially underestimated people's accuracy about their attribute weights: The decision scientists, on average, predicted a correlation between reported and best-fit weights of 0.44, compared to the average observed correlation in Studies 1A-1B of 0.80 and 0.86 (respectively). At the same time, the decision scientists did accurately predict the percent of people who reported their choice method correctly, suggesting that our results serve to confirm some beliefs in the field while challenging others. These new data provide a clearer benchmark of the field's prior expectations, against which we can compare our empirical results. We now report these data as Study 4 in the revised manuscript (p. 12-13), and hope that they help further situate our experiments within the ongoing debate about introspective accuracy about choice processes.

Another prominent issue was the ecological validity of the tasks. These tasks are similar to many multi attribute decisions in the Judgment and Decision Making & Marketing literatures. There is clearly value in studying these kinds of tasks, if only to address those literatures. It does seem worth noting that the amount of introspection could be very different in other more natural settings. Presenting information in tables allows decision makers to easily screen out attributes that they want to ignore. But when attributes are not nicely separated and organized it may be difficult to ignore those attributes. For example, when purchasing a house, ugly colored walls may make you hate the house, even knowing that you could easily repaint them and so should ignore that attribute. It is simply not possible to avoid looking at the walls when walking through a house. The current study cannot address examples like this one.

We agree with this concern. In the revised manuscript, we have moved discussion of this limitation to the top of the "Limitations of the present studies" section in the General Discussion (p. 17), and emphasize the need for future research to test introspective accuracy in more realistic settings with implicitly-presented attributes. We have also further emphasized this limitation in other sections of the General Discussion. It is worth noting that we are already working on additional and more naturalistic versions of this task to further understand this.

This gets me to the lingering concern that I mentioned in my summary paragraph. With so many attributes, it is quite likely, and consistent with Figure 2A, that people would ignore several attributes, i.e., assign them weights of zero. That would be a very conscious decision to not look at several attributes, in order to simplify these choices. The question is whether this accounts for a substantial amount of the correlation between fitted and reported weights. There are a couple of ways the authors could address this. First, they could use their "information gain" metric but exclude reported weights of 0 when computing the metric. Second, they could examine how often the reported rankings of the top two or three attributes aligns with the fitted rankings of those same attributes. This would focus the analysis on what matters, the weighting of the most important attributes, and not on the irrelevant attributes.

We agree that this is an important concern, and we now address it using the analytic approach the Reviewer suggested. First, we redid all the weight accuracy analyses, dropping any attributes whose reported weight had an absolute value less than 0.05 (to be conservative). We found that, as when dropping participants who only used a single attribute, participants' weight accuracy went down slightly but still remained very high. Concretely, in Study 1A, the average participant-level correlation between reported and fitted weights dropped from 0.80 to 0.75, and the percent of information in the fitted weights contained in the reported weights dropped from

79% to 77% – a minimal drop in both cases. The patterns in other studies were similar. Together, these results indicate that the high weight accuracy observed in our studies is not primarily due to zero or near-zero weights. (Weight accuracy goes down by even less if only dropping the reported weights that were literally zero, as the Reviewer suggested.) These analyses are reported on p. 6 of the revised manuscript.

Second, we again redid the weight accuracy analyses, this time only including for each participant the three attributes they had reported as having the highest absolute weight. Again, we found that, with this restriction, participants' weight accuracy went down slightly but still remained very high. Concretely, in Study 1A, the average participant-level correlation dropped from 0.80 to 0.71, and the percent of information in the fitted weights contained in the reported weights actually increased from 79% to 82%. The patterns in other studies were similar. These analyses are reported on p. 6 of the revised manuscript.

Together, these results indicate that the observed high level of weight accuracy was not primarily due to including less important or ignored attributes.

Reviewer #5 (Remarks to the Author):

I was given this manuscript, with its peer-review history, to evaluate the quality of the rebuttal, notably regarding the issues and concerns raised by Reviewers 1 and 3.

In my view, the main reservations of both reviewers concerned two main aspects: 1) the initial framing of the study, which they viewed as an exaggerated refutation of a strawman argument (namely “researchers believe that individuals have no access to their decision criteria”); and 2) the ecological validity of the task, which, by systematically varying many attributes of options over hundreds of trials with limited stakes, could lack generalizability and might incidentally improve introspective accuracy.

I fully agree with the rationale of these initial reservations, yet consider (like reviewer 3) that the manuscript was nonetheless based on a solid methodology (task design, replications, solid sample size, advanced/innovative analytical approach, etc.) and contains material, which, even under a more reasonable framing, could advance the debate on human introspective ability about their decision motives and be of (methodological) interest to a relatively large crowd of researchers.

In their revision, I have the feeling that the authors have seriously amended their manuscript to address most concerns, and reframed their contribution/angle in a much more reasonable and productive direction. I also think some reservations raised by Reviewer 1 were overly critical and required the present work to raise to uncommon standards in behavioral sciences: sure, this work is not perfect and some methodological comprises had to be devised, and sure this work cannot settle the debate once and for all; but, again, I think in its revised form, it constitutes a reasonable, more nuanced, and solid contribution to the scientific question at hand.

We thank the Reviewer for stepping in to complete the review, and for their encouraging feedback.

Here are a couple of additional (minor) thoughts:

1) I have the feeling that the results are framed in a way that suggest that we evaluate the “true” weights – i.e. motives/preferences- of participants. I have two reservations about this: first, the task is barely value-based (the stakes are pretty low in the context of the hypothetical study 1 and the incentive study 2: watch a movie trailer), so it should be made explicit that those weights might very-well be quite idiosyncratic to the task. Second, and in the same line, there might very well be an interaction between the task design (i.e. the distributions from which are drawn the attribute values) and participants actual valuation of the attributes. For instance, there are some models in multi-attribute choice, that weight attributes as a function of their relative variance in the choice set (because it becomes the main differentiation characteristics). This possibility of context-dependency / idiosyncrasy of the weights should be discussed.

We agree with these points and with the likely possibility of participants forming task-idiosyncratic or context-dependent weights. By “true” weights, we had meant to indicate the weights that the participant actually used in the context of this study – not necessarily the values that participants would actually place on these attributes in analogous real-life choices. We have now amended the manuscript to emphasize that the weights guiding participants’ choices in these tasks may be very different from their corresponding attribute values in real life. Most directly, in the limitations section of the General Discussion (p. 17), we now say: “Moreover, participants’ choice processes in these artificial contexts may have a limited correspondence to the processes people would use in real-life versions of these choices – e.g., the attribute weights employed in these contexts may not be representative of the weights people would actually place on analogous attributes in real life. A key future direction...will be to probe accuracy in more realistic choice formats.”

2) Related to this, I would expect to see the distribution of weights splitted per attributes (could be in the SOM) to see how variable/consistent are participants relative weighting of the different attributes. Would there be a way to improve the Figure 2A to include this ? (also because the statistical model used in the Main text do use random intercepts and slopes per individual and attribute).

We appreciate this suggestion and have created these graphs. Unfortunately, we were unable to include them in Figure 2A (without distracting from the main results), and now report these graphs in the Supplemental Information (see SI 3.4). Here is the graph for Study 1A:

The graph shows two overlapping histograms for each attribute: the reported weights in blue, and the best-fit weights in beige (with the light maroon indicating the overlap). The attributes all had somewhat similar distributions of weights; all had a mode at zero and a standard deviation between roughly 0.2 to 0.4. Some also had a mode at 1; others had a more uniform distribution across the rest of the range. And some were, of course, weighted more heavily than others on average.

3) An important interpretational dimension, also, is that introspection is made retrospectively on choices. This means that this manuscript does not investigate participants general introspective accuracy about their preferences/values and its link to choices (e.g. in the future), but a more limited version of introspection accuracy, which tries to rationalize, ex-post, a set of choices. I think the authors should pay more attention to this dimension, to nuance their angle and findings. E.g. in the abstract, a liberal use of the “general” present tense (e.g. “How much awareness do people have about how they make such choices?”) could be replaced with a past tense (e.g. “How much awareness do people have about how choices they have made?”). This type of small modification could/should be generalized throughout the abstract and manuscript, and the subtlety briefly discussed in the discussion.

We have modified the language throughout the manuscript to make it clear that our experiments examine introspective accuracy about choice processes used in the past, as reported retrospectively. We also now raise this point explicitly in the “Limitations” section of the General Discussion (p. 18).